

# Neonatal hearing screening among high-risk newborns in Northwestern Nigeria

Semen Stephen Yikawe[1], Nasir Aliyu[2], Joseph Hassan Solomon[3], Shuaibu Lawal[2], Musbau Adeyemi[2], Agatha Eileen Wapmuk[4], Andrew Musa Adamu[5,6], Anayochukwu Edward Anyasodor[7] and Oyelola A. Adegboye[8]

[1] Nigerian Air Force Hospital, Ikeja, Lagos, Nigeria
[2] Federal Teaching Hospital, Katsina, Katsina, Nigeria
[3] Nigerian Airforce Hospital, Abuja, Federal Capital Territory, Nigeria
[4] Nigerian Institute for Medical Research, Lagos, Lagos, Nigeria
[5] Australian Institute of Tropical Health and Medicine, James Cook University, Townsville, Queensland, Australia
[6] College of Public Health, Medical and Veterinary Sciences, James Cook University, Townsville, Queensland, Australia
[7] Rural Health Research Institute, Charles Sturt University, Orange, NSW, Australia
[8] Menzies School of Health Research, Charles Darwin University, Darwin, NT, Australia

Corresponding author
Oyelola A. Adegboye,
oyelola.adegboye@menzies.edu.au

## ABSTRACT

**Background.** Neonatal Intensive Care Unit (NICU) patients are at an increased risk of developing hearing loss. Given the critical role of normal hearing in speech and language development, early detection and treatment of this condition in children are paramount.

**Methods.** Hearing assessments were performed using transient evoked otoacoustic emission (TEOAE) and automated auditory brain stem response (AABR) tests on 60 neonates. The resulting data were analysed using R version 4.3.3.

**Results.** Out of the 60 neonates enrolled in the study, 57% were males and 43% were females, and their ages ranged from 1 to 30 days. A total of 43 (71.7%) neonates passed both hearing tests, while 17 (28.3%) failed. In the unadjusted analysis, low birth weight, prematurity, birth asphyxia, and gestational age were significantly associated with neonatal hearing screening failure. Premature neonates and those with low birth weight had markedly higher odds of failing the screening (Odds Ratio (OR) = 24.4; 95% confidence interval (CI) [5.85–135.0] and OR = 12.3; 95% CI [3.5–50.6], respectively), while gestational age was associated with lower odds of failure (OR = 0.59; 95% CI [0.43–0.76]). In the multivariable model, after accounting for multicollinearity, only gestational age remained a statistically significant predictor, with each additional week associated with reduced odds of screening failure (adjusted OR = 0.60; 95% CI [0.43–0.80]).

**Conclusion.** Our findings underscore the importance of early screening for hearing loss in high-risk neonates in the NICU to facilitate timely interventions.

## INTRODUCTION

Congenital hearing loss affects 2–4% of Neonatal Intensive Care Unit (NICU) neonates globally, with higher burdens in lower and middle-income countries (LMICs) due to preventable factors like perinatal asphyxia and untreated maternal infections (*Olusanya et al., 2020*). Approximately 5 per 1,000 neonates in LMICs have congenital or early childhood onset sensorineural hearing loss (SNHL), with higher prevalence rates observed in NICU-admitted infants (2–4%) (*Olusanya et al., 2020*). Up to 50% of these cases could be prevented or mitigated through early detection, prompt diagnosis, and effective management strategies (*Ullauri et al., 2014*). Hearing loss in infants is particularly worrisome because normal hearing is essential for speech and language development at a critical stage in the child's growth (*Al-Ani, 2023*; *Choe, Park & Kim, 2023*). This condition eventually affects the child's emotional, psychological and general well-being (*Adily et al., 2024*; *Gazia et al., 2019*; *Timmer et al., 2024*). Therefore, early diagnosis and treatment are crucial to enabling hearing-impaired infants to develop speech and achieve parity with their peers. To benefit from normal speech and language development, a diagnosis of hearing loss in infants should be made before three months of age, and treatment should be instituted before six months of age (*Joint Committee on Infant Hearing, 2019*). Universal screening of all infants is encouraged for early diagnosis, which precedes the commencement of prompt hearing rehabilitation for speech and language development. This screening targets two categories of infants: well-babies and those admitted in NICU (*Labaeka et al., 2018*). The recommended method currently used for newborn hearing screening (NHS) is otoacoustic emission (OAE) and/or automated auditory brainstem response (AABR) (*Akinpelu et al., 2014*; *Samaddar et al., 2015*).

Neonates admitted to the NICU account for 4–8% of all live births (*Busa et al., 2007*; *Chang, Oh & Park, 2020*; *WHO, 2010*). It is estimated that 0.5 to 5 per 1,000 neonates have congenital or early childhood onset SNHL (*WHO, 2016*; *WHO, 2010*). However, the prevalence of hearing loss among neonates admitted to the NICU is much higher (2–4%) (*Arslan et al., 2013*). A study in South Korea compared hearing screening outcomes in well-babies and patients admitted to NICU and observed an average prevalence of 5.6/1000 live births, with the latter group showing a higher prevalence of hearing loss (*Chang, Oh & Park, 2020*). Hearing loss among neonates has been documented, with a study reporting a prevalence of 5.5 per 1,000 live births among those referred to AABR testing and 22 per 1,000 live births among patients admitted to special care baby unit (SCBU) (*Olusanya, Wirz & Luxon, 2008*). Additionally, neonates with identified risk had a 9.5% prevalence of hearing loss, with 95% experiencing bilateral hearing impairment (*Labaeka et al., 2018*).

In Nigeria, over the past two decades, improvements in neonatal healthcare systems, such as expanded NICU coverage and better-equipped centres, have led to the proliferation of neonatal intensive care services, highlighting the need for routine hearing assessment in NICU-admitted neonates for early diagnosis and timely intervention. Despite the improvements, no national data exist on newborn hearing loss, and the prevalence is likely higher in Nigeria due to gaps in maternal and child healthcare services and weaker health systems, particularly in the rural and suburban areas. While similar studies have been

conducted in other regions of Nigeria, Northwestern Nigeria, a highly populated region with significant maternal and child health challenges, remains largely understudied (*Labaeka et al., 2018*; *Olarewaju, 2021*; *Oyinwola et al., 2023*). For instance, a study conducted in Southwest Nigeria reported a failed neonatal hearing screening of 41.5% at admission and 15.9% at discharge (*Labaeka et al., 2018*). Furthermore, a study on neonatal hearing screening (AABR) on neonates diagnosed with hyperbilirubinaemia reported a failed neonatal hearing screening of 26.2% (*Oyinwola et al., 2023*). These rates are significantly higher than those reported from developed countries. Given that maternal and child health is a key priority of the Sustainable Development Goals (SDG 3), which promotes well-being for all ages (*United Nations, 2023*), the early identification of neonatal hearing loss and understanding its risk factors in Northwestern Nigeria is critical. Therefore, this study aims to assess the rate of hearing screening failure among high-risk neonates in Northwestern Nigeria, contributing to local epidemiological evidence to inform early screening strategies and guide healthcare policy interventions.

## MATERIALS AND METHODS

### Study design, location and participants

This study employed a cross-sectional design. The study was conducted from October 2023 to March 2024 at the NICU of the FTH Katsina. FTH Katsina is a key tertiary health facility serving Katsina State in Northwestern Nigeria and neighbouring states and countries. The Ear, Nose, and Throat department offers comprehensive audiological services, while the hospital's Paediatric department includes a well-equipped NICU with a capacity of 35 beds and an average weekly intake of 15 to 20 neonates. Eligibility criteria included all neonates admitted into the NICU whose guardians consented to participate. Neonates with congenital ear deformities or craniofacial anomalies were excluded to minimise confounding in OAE measurements. Enrolment was consecutive, ensuring a representative sample of the NICU population.

### Data collection methodology

Data were collected using a semi-structured questionnaire. The tool was pretested for clarity and clinical relevance, demonstrating good internal consistency and inter-rater reliability. It captured maternal health history, perinatal complications, and neonatal admission diagnoses. Clinical assessments using specialised audiological equipment were also carried out. Specifically, the HEINE Otoscope (model mini-3000; HEINE Optotechnik GmbH & Co., Munich, Germany) and the Intelligent Hearing System OAE/AABR Machine (Intelligent Hearing Systems, Miami, FL, USA) were used for ear examinations and hearing tests. These tools were calibrated according to the manufacturer's guidelines to ensure accuracy in the neonatal hearing screening. The data collected were entered into Excel sheets before data analysis in R version 4.3.3 (*R Core Team, 2024*).

#### Transient evoked otoacoustic emission procedure

The initial visual inspection of the ear canal and tympanic membrane was followed by transient evoked otoacoustic emission (TEOAE) tests using the Intelligent Hearing System

device. The procedure involved presenting transient sounds and recording the outer hair cell responses, with the system automatically determining test outcomes "pass"/"fail" based on the preset binomial statistical probability that an emission has been recorded within the frequency range 1.5–3.5 kHz. The tester/screener could not alter the default settings but could decide to repeat the recording based on qualitative information about stimulus stability and artefacts provided by the instrument (*Liu & Hatzinakos, 2014*; *Satish, Anil Kumar & Viswanatha, 2019*; *Wu & Liu, 2017*). A typical recording took an average of about 3 min.

### Automated auditory brainstem response procedure

The AABR test, conducted in a controlled environment, assessed the severity of hearing loss across different frequencies. The screening outcomes were based on statistical confidence levels in signal detection, categorised into "pass" or "refer" for further evaluation. The screening was conducted in a quiet room while the baby was calm or sleeping; both ears were screened simultaneously at frequencies of 35, 40, and 70 dB. The screener automatically displayed "pass" when it had collected sufficient data to establish, with 99.96% statistical confidence, that an auditory brainstem response (ABR) signal was present and consistent with the template at a minimum of 1,000 sweeps when screening at 35 dB clicks and a minimum of 2,000 sweeps when screening at 40 dB and 70 dB clicks. "Refer" result was displayed when it did not establish with 99.96% statistical confidence that the ABR signal was present at 15,000 sweeps when screening at 35- and 40-dB clicks or at 10,000 sweeps when screening at 70 dB clicks (*Boudewyns et al., 2016*; *Heidari, Manesh & Rajabi, 2015*; *Oyinwola et al., 2023*; *Satish, Anil Kumar & Viswanatha, 2019*).

## Variables
### Outcome variables

The primary outcome was the results of the hearing screening, assessed using TEOAE and AABR. This outcome was treated as a binary variable: "Fail" = 1 and "Pass" = 0. Neonates were deemed to have passed the hearing screening if they passed both the TEOAE and AABR tests or failed the TEOAE but passed the AABR. In accordance with international guidelines (*American Speech-Language-Hearing Association, 2023*; *WHO, 2021*), any neonate who failed the AABR was classified as having failed the hearing screening, regardless of their TEOAE result, and was referred for a full audiological evaluation.

### Independent variables

The independent variables included a range of maternal and neonatal characteristics. Gender was recorded as Male or Female. Age in days was captured as a continuous variable reflecting the neonate's age at the time of screening. Perinatal maturity and growth indicators such as gestational age (GA) at birth (in weeks) was treated as a continuous variable and used to derive prematurity, defined as birth before 37 completed weeks, while birth weight was recorded in grams as a continuous variable and categorised as low birth weight (LBW) if less than 2,500 grams. Mode of delivery was categorised as spontaneous vaginal delivery or caesarean section. Neonatal sepsis was documented as a binary variable (Yes/No), based on clinical diagnosis during admission to the NICU. Birth asphyxia was

defined as any difficulty in initiating breathing immediately after birth. Other binary variables included infection during pregnancy, pregnancy-related co-morbidities, and ototoxic drug use during pregnancy, based on maternal report. Mother's age was recorded as a continuous variable in years.

## Sample size calculation

Based on a prevalence of hearing loss ranging from 2% to 4% (*Busa et al., 2007*; *Hille et al., 2007*), we calculated a required sample size of 31 to 60 participants to achieve a 95% confidence interval with a precision of 5%. Despite this, data were collected from 60 patients.

## Data analysis

Frequency distributions and cross-tabulations were carried out to examine the relationships between variables. A chi-square test or Fisher's exact test was utilised to assess associations between categorical variables, while the Wilcoxon rank sum test was employed to evaluate differences in continuous variables across groups. The proportion of neonates who failed hearing screening was calculated using Clopper–Pearson's 95% confidence interval (*Clopper & Pearson, 1934*). Univariate logistic regression was initially performed to assess the crude association between each independent variable and hearing screening failure. Moreover, a multivariable logistic regression model was utilised to adjust for potential confounding factors and identify independent predictors of hearing screening failure. Results were presented as odds ratio (OR) and adjusted odds ratio (aOR) with their 95% confidence interval (CI) for univariate and multivariable logistic models, respectively.

To evaluate multicollinearity among predictors included in the multivariable logistic regression model, the variance inflation factor (VIF) was calculated for each covariate. VIF values exceeding 5 are generally indicative of moderate multicollinearity, while values above 10 are considered indicative of severe multicollinearity (Table S1). In our analysis, most variables showed acceptable VIFs, except for birth weight (VIF = 15.53) and GA (VIF = 9.76), suggesting potential collinearity concerns.

To further explore these relationships, we computed a correlation matrix for four closely related variables: GA, birth weight, LBW, and premature (Fig. S1). The results revealed strong correlations between GA and premature ($r = -0.79$), and between birth weight and LBW ($r = -0.78$), indicating overlapping information. Furthermore, LBW and premature were highly correlated ($r = 0.81$), which may introduce redundancy in multivariable models. Therefore, the multivariable logistic regression only includes GA among the four variables.

All analyses were performed using R version 4.3.3 (*R Core Team, 2024*), with statistical significance assessed at a 5% alpha level.

## Ethics

Ethical approval was obtained from the Ethics and Research Committee of Federal Teaching Hospital, Katsina (Approval number: FTHKTNHREC.REG.24/06/22C/101). Written informed consent was obtained from the parents or guardians of the babies before they

**Table 1  Patients clinical profile.**

| Characteristic | Overall | Fail, *N* = 17 | Pass, *N* = 43 | *p*-value[a] |
|---|---|---|---|---|
| Hearing screening failure, % (95% CI) [b] | 60 | 28.3% (17.5 to 41.4) | | |
| Gender, n (%) | | | | 0.35 |
| Male | 34 (57) | 8 (47) | 26 (60) | |
| Female | 26 (43) | 9 (53) | 17 (40) | |
| Age in days, Median (Q1, Q3) | 11 (6, 18) | 16 (8, 20) | 10 (6, 17) | 0.33 |
| Gestational age in weeks, Median (Q1, Q3) | 37 (34.5, 39) | 34 (33, 36) | 38 (36, 40) | **<0.001** |
| Birth weight (grams), Median (Q1, Q3) | 2500 (200, 2800) | 1900 (1800, 2200) | 2600 (2300, 2800) | **0.011** |
| Mothers' Age (Years), Median (Q1, Q3) | 28 (24, 31) | 29 (25, 32) | 28 (23, 31) | 0.26 |
| Delivery, n (%) | | | | 0.51 |
| Spontaneous vaginal delivery | 46 (77) | 12 (71) | 34 (79) | |
| Caesarean section | 14 (23) | 5 (29) | 9 (21) | |
| Neonatal sepsis, n (%) | 21 (35) | 4 (24) | 17 (40) | 0.24 |
| Birth asphyxia, n (%) | 13 (22) | 8 (47) | 5 (12) | **0.005** |
| Low birth weight, n (%) | 19 (32) | 12 (71) | 7 (16) | **<0.001** |
| Premature, n (%) | 14 (23) | 11 (65) | 3 (7.0) | **<0.001** |
| Infection during pregnancy, n (%) | 7 (12) | 5 (29) | 2 (4.7) | **0.016** |
| Co-Morbidities during pregnancy, n (%) | 5 (8.3) | 2 (12) | 3 (7.0) | 0.62 |
| Ototoxic drug use during pregnancy, n (%) | 2 (3.3) | 2 (12) | 0 (0) | 0.077 |

Notes.
[a]Pearson's Chi-squared test; Wilcoxon rank sum test; Fisher's exact test.
[b]Clopper–Pearson confidence interval (CI), Q1, Q3, First and third quartile; Bold *p*-values are significant.

were enrolled in the study. Participant anonymity was strictly maintained to protect confidentiality.

## RESULTS

A total of 60 babies were enrolled in this study, comprising 34 (57%) males and 26 (43%) females (Table 1). Participants' ages ranged from 1 to 30 days, with a median age of 11 days (Inter-quartile range: 6, 18). The gestational age ranged from 26 to 41 weeks, with a median of 37 weeks (IQR = 34.5, 39). The median birth weight was 2.5 kg (IQR: 2, 2.8). The distribution of gender, mode of delivery and hearing test results, broken down by birth weight and gestational age, is presented in Fig. 1. The figure showed a pattern of slightly more males at higher birth weights and gestational categories clustered around the median (Figs. 1A and 1B). A higher proportion of the neonates passed the hearing test at higher birth weights and gestational age categories (Figs. 1C and 1D). Although spontaneous vaginal delivery is common across all birth weights, we observed no difference across birth weight and gestational categories (Figs. 1E and 1F).

All 60 participants underwent OAE and AABR testing, 43 (71.7%) passed, while 17 (28.3%) failed, yielding a hearing screening failure prevalence of 28.3% (95% CI [17.5–41.4]) among NICU-admitted neonates (Table 1). Clinical factors that showed significant associations with failure of screening tests were gestational age, birth weight and prematurity. Neonates who failed the screening tests had a significantly lower median

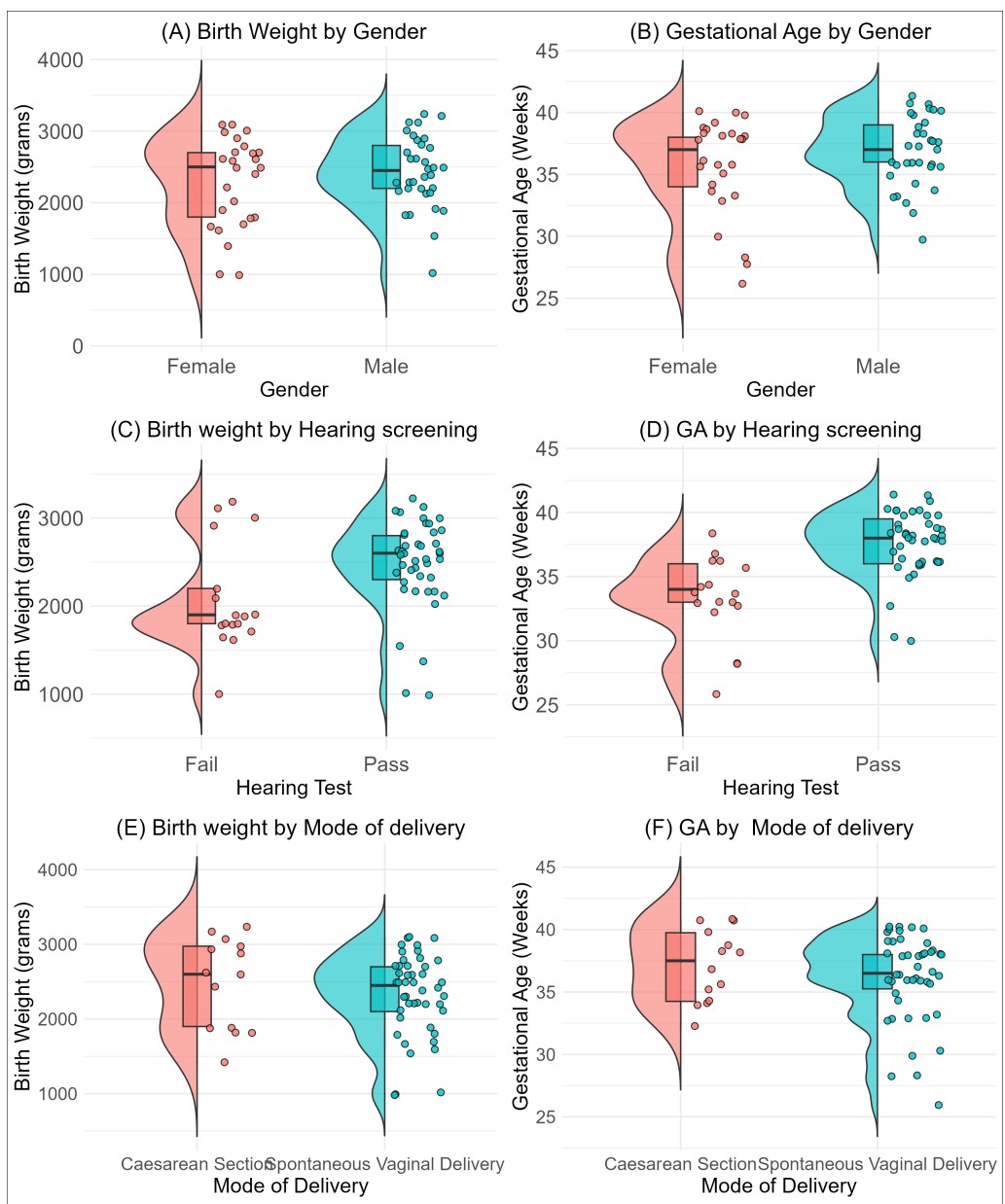

**Figure 1** Distribution of birth weight (B, D, F) and gestational age (A, C, E) stratified by neonatal characteristics. (A, B) show distributions by neonates gender, (C, D) by hearing test outcome (pass/-fail), and (E, F) by mode of delivery (Caesarean section *vs.* spontaneous vaginal delivery).

gestational age (34 weeks *vs* 38 weeks, *p* < 0.001). Similarly, the median weight of neonates who failed the screening tests was significantly lower, 1.9 kg, than that of those who passed the test, 2.6 kg, *p* = 0.011. In the case of prematurity, a substantial 65% of preterm neonates failed the screening tests compared to only 7% of term neonates (*p* < 0.001). Other factors like neonatal sepsis, birth asphyxia, and infections during pregnancy were also explored, with birth asphyxia and infection during pregnancy showing a significant

association with failure of screening tests ($p < 0.05$). However, maternal factors such as age and co-morbidities during pregnancy, as well as the method of delivery, did not show a significant impact on the screening test outcomes.

Table 2 presents both unadjusted and adjusted analyses of potential risk factors for hearing screening failure. In the unadjusted analysis, gestational age, LBW, prematurity, birth weight and birth asphyxia showed statistically significant associations. Neonates born prematurely and those with low birth weight had the highest likelihood of hearing screening failure compared to those born at term (OR = 24.4; 95% CI [5.85–135.0]) and those with normal birth weight (OR = 12.3; 95% CI [3.5–50.6]). In contrast, higher gestational age was significantly associated with lower odds of failing the hearing screening (OR = 0.59; 95% CI [0.43–0.76]), suggesting that more matured neonates were more likely to pass the test.

In the multivariable model, due to multicollinearity, only gestational age was included among the perinatal maturity and growth indicators. After adjusting for potential confounders, the associations observed in the univariable analyses were no longer statistically significant, except for gestational age. Higher gestational age was significantly associated with lower odds of failing the hearing screening (OR = 0.60; 95% CI [0.43–0.80]), indicating that each unit (week) increase in gestational age was associated with a 40% reduction in the odds of failing the test.

## DISCUSSION

This study investigated the rate of neonatal hearing screening failure (refer rate) among high-risk neonates in Northwestern Nigeria. Our results indicate that 28.3% of 60 neonates failed the hearing tests after assessment. This report is similar to a study in Bangladesh, where a failure rate of 23% was observed (*Shameem et al., 2022*). In another study in Ibadan, Nigeria, an even higher failure rate of 41% (OAE test on admission) was reported among newborns, which dropped to 15.9% during a repeat OAE test at discharge (*Labaeka et al., 2018*). In developed countries, neonates exhibit much lower rates of hearing screening failure. For instance, Italy and the United Kingdom recorded failure rates of 9.1% (*Ciorba et al., 2019*) and 2.6% (*Wood, Sutton & Davis, 2015*), respectively. The observed increased failure rate among neonates in developing countries stems from preventable factors such as limited access to healthcare, low uptake of immunisation, infections like rubella and weak health systems. A higher proportion of male neonates undergoing screening (57% *vs.* 43%) likely reflects two factors. Firstly, the elevated male-to-female ratio (1.3:1) is consistent with global patterns of higher male admissions due to biological vulnerabilities (*e.g.*, respiratory distress syndrome) (*Peacock et al., 2012*). Secondly, cultural preferences in Northern Nigeria, where male infants may be prioritised for healthcare access during resource constraints, could also be a factor. However, gender was not significantly associated with screening failure rates, suggesting that test performance was equitable among those screened.

Our univariable analysis identified several perinatal factors, including birth weight, gestational age, and prematurity, as significant predictors of newborn hearing screening

**Table 2** Associated risk factors of hearing screening failure.

| Characteristic | OR (95% CI) | *p*-value | aOR (95% CI) | *p*-value |
|---|---|---|---|---|
| Gender | | | | |
| Female | Ref | | Ref | |
| Male | 0.58 (0.18, 1.81) | 0.347 | 0.56 (0.08, 3.40) | 0.52 |
| Age in days | 1.04 (0.97, 1.12) | 0.301 | 1.03 (0.91, 1.16) | 0.6 |
| Gestational age (Weeks) | **0.59 (0.42, 0.76)** | **<0.001** | **0.61 (0.43, 0.80)** | **0.002** |
| Birth weight[a] (grams) | **1.00 (1.00, 1.00)** | **0.019** | Not included | |
| Low birth weight | | | | |
| No | **Ref** | | Not included | |
| Yes | **12.30 (3.50, 50.60)** | **<0.001** | | |
| Mother's age (Years) | 1.08 (0.96, 1.22) | 0.222 | 1.04 (0.86, 1.29) | 0.67 |
| Delivery | | | | |
| Caesarean section | Ref | | Ref | |
| Spontaneous vaginal delivery | 0.635 (0.18, 2.41) | 0.486 | 0.84 (0.11, 8.01) | 0.87 |
| Neonatal sepsis | | | | |
| No | Ref | | Ref | |
| Yes | 0.47 (0.12, 1.59) | 0.247 | 0.39 (0.04, 2.98) | 0.38 |
| Birth asphyxia | | | | |
| No | Ref | | Ref | |
| Yes | **6.76 (1.84, 27.50)** | **0.005** | 3.92 (0.56, 29.1) | 0.16 |
| Premature | | | | |
| No | Ref | | Not included | |
| Yes | **24.40 (5.85, 135.00)** | **<0.001** | | |
| Infection during pregnancy | | | | |
| No | Ref | | Ref | |
| Yes | **8.54 (1.62, 65.20)** | **0.017** | 10.0 (0.73, 282) | 0.11 |
| Co-morbidities during pregnancy | | | | |
| No | Ref | | Ref | |
| Yes | 1.78 (0.22, 11.80) | 0.550 | 0.71 (0.02, 13.7) | 0.83 |

**Notes.**
Confidence Interval (CI); Odds Ratio (OR).
[a]Codds change per 10 g increase in birthweight.
Bold font indicates statistical significance.

outcomes, highlighting a potential relationship with auditory function (*Savenko, Garbaruk & Krasovskaya, 2020*). In the multivariable model, gestational age remained independently associated with screening failure, reinforcing its role as a key determinant. This aligns with prior studies showing that prematurity is a significant risk factor for hearing impairment (*Wallois, Routier & Bourel-Ponchel, 2020*; *Wroblewska-Seniuk et al., 2018*), with the risk increasing as gestational maturity decreases (*Wroblewska-Seniuk et al., 2017*). The preceding emphasises the susceptibility of preterm infants to hearing loss and reinforces the importance of prompt intervention to mitigate exacerbation of the condition (*Al-Ani, 2023*; *Sizer, Muluk & Ankle, 2023*). In addition, other identified significant risk factors included maternal infection during pregnancy and birth asphyxia, which have been previously reported (*Savenko, Garbaruk & Krasovskaya, 2020*). This necessitates preventive

approaches, including monitoring during antenatal care (ANC), to reduce the risk of hearing impairment in neonates (*Holzinger, Fellinger & Hofer, 2022*).

The report from our study highlights the urgent need for interventional strategies to address the potential underlying factors that contribute to adverse birth outcomes. This will involve screening programmes to identify victims of hearing loss, targeting neonates with lower birth weight and/or gestational age, as they are associated with a higher risk of failure of neonatal hearing screening (Table 2). Our observations underscore the importance of ANC and the promotion of maternal health for achieving optimal birth outcomes. This includes ensuring maternal and child health services, as outlined in Sustainable Development Goal 3 (*UNICEF, 2023*).

A key limitation of this study is the absence of follow-up diagnostic confirmation for neonates who failed the initial screening. As a result, we were unable to assess the accuracy of the screening process, including sensitivity, specificity, or predictive values. Also, despite the global importance of neonatal hearing screening, our study was limited by a relatively small sample size, primarily due to parental reluctance to participate for cultural reasons. These parents also withheld their consent for the neonates to be enrolled. However, the research highlights the importance of hearing screening and the risk factors associated with hearing loss among babies admitted to the NICU. Elucidating the risk factors of hearing loss among high-risk infants and promptly identifying and treating neonates with hearing impairment could reduce the burden of childhood hearing loss in the region. Overall, the small sample size and lack of diagnostic follow-up limit generalizability. Future studies could incorporate longitudinal designs with follow-up diagnostic ABRs to confirm the prevalence of hearing loss.

## CONCLUSION

This study highlights a high newborn hearing screening referral rate and identifies key perinatal risk factors that warrant targeted screening strategies. The significantly higher failure rates observed among neonates in developing countries underscore the influence of preventable factors such as inadequate healthcare access, suboptimal immunisation coverage, and systemic health disparities. Importantly, risk factors such as gestational age, low birth weight, prematurity, and maternal health issues like infections during pregnancy have been identified as significant predictors of neonatal hearing screening failure. These findings underscore the crucial need for targeted interventional strategies encompassing comprehensive neonatal hearing screening programs, enhanced antenatal care, and promoting maternal health to mitigate these risks. The urgency for such interventions aligns with global health priorities, emphasising the importance of neonatal and maternal health services to achieve optimal birth outcomes.

### Funding

The authors received no funding for this work.

## Competing Interests

Oyelola A. Adegboye is an Academic Editor for PeerJ.

## Author Contributions

- Semen Stephen Yikawe conceived and designed the experiments, performed the experiments, analyzed the data, authored or reviewed drafts of the article, and approved the final draft.
- Nasir Aliyu performed the experiments, authored or reviewed drafts of the article, and approved the final draft.
- Joseph Hassan Solomon performed the experiments, authored or reviewed drafts of the article, and approved the final draft.
- Shuaibu Lawal performed the experiments, authored or reviewed drafts of the article, and approved the final draft.
- Musbau Adeyemi performed the experiments, authored or reviewed drafts of the article, and approved the final draft.
- Agatha Eileen Wapmuk performed the experiments, authored or reviewed drafts of the article, and approved the final draft.
- Andrew Musa Adamu performed the experiments, authored or reviewed drafts of the article, and approved the final draft.
- Anayochukwu Edward Anyasodor performed the experiments, authored or reviewed drafts of the article, and approved the final draft.
- Oyelola A. Adegboye analyzed the data, prepared figures and/or tables, and approved the final draft.

## Human Ethics

The following information was supplied relating to ethical approvals (i.e., approving body and any reference numbers):

The Research Committee of Federal Teaching Hospital, Katsina, approved the study (Approval Number: FTHKTNHREC.REG.24/06/22C/101).

## Data Availability

The raw data is available in the Supplemental Files.

## Supplemental Information

Supplemental information for this article can be found online at http://dx.doi.org/10.7717/peerj.20002#supplemental-information.

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
