# Peer review of "Neonatal hearing screening among high-risk newborns in Northwestern Nigeria"

_PeerJ, doi:10.7717/peerj.20002_

## Round 0.1 · original submission · Major Revisions

Please follow the reviewers' advice.

·

Basic reporting

Regarding the structure of the manuscript, the Conclusion section seems to be missing.

The authors provided one figure that is difficult to read and would benefit from improvements (see additional comments).

Experimental design

The methods were not described with sufficient detail & information to allow replication (see also additional comments). The study also did not answer the study question, as the aim of the study and the title of the manuscript are misleading.

Validity of the findings

All underlying data are provided.
However, the conclusions are not fully related to the original research question, and some concerns have been raised about the regression analysis.

Additional comments

Thank you for the opportunity to review this interesting work entitled “Early detection of hearing loss in neonatal intensive care unit: prevalence, risk factors and screening accuracy in Northwestern Nigeria”.

There are some comments I would like to make to improve the quality of the manuscript.

1) The title suggests that the authors will report the prevalence of hearing loss, risk factors for hearing loss, and screening accuracy found in their study.
I have comments on all three 3 factors.
Prevalence: I do not believe that the prevalence of hearing loss in this study was as high as 28.33%, as the authors state on lines 170-171. To my understanding, this is the percentage of neonates who failed the screening and were referred to further evaluation regarding hearing loss (see lines 138-140). There is no mention of the outcome of the audiological evaluation in the manuscript, which would confirm in which neonates a hearing loss was diagnosed. In the introduction (lines 70-71), the authors refer to a study in which the hearing loss of neonates admitted to the NICU was approximately 2-4%. It would be very unlikely that the prevalence in the present study would be as high as 28%.
Risk factors for hearing loss: Since no results for hearing loss were presented, no risk factors could have been identified.
Screening accuracy: No results regarding screening accuracy were presented in the manuscript.
I would recommend adding those factors to the manuscript or changing the name of the title.

2) Abstract:
- You state that 44 neonates passed and 15 failed the screening. Later in the manuscript, those numbers are 43 and 17 (for example, see Table 1). Please clarify which numbers are correct.
- The odds ratios in the abstract and their relation to hearing loss should be checked, see my later comments.

3) Introduction
- Lines 89-92: The aim of the study is to determine the prevalence and risk factors for hearing loss. As stated previously, both factors seem not to be assessed in the study.

4) Materials and methods
- Chapter 2.1. It would be good to add the dates of the study to this section. I understand that 60 neonates were enrolled, but it would be good to know the time period of the study.
- Chapter 2.1: I am wondering what the participation rate in the study is. What percentage of all potential candidates is the sample of 60 neonates? The information would allow us to make a picture of how representative the sample is. Do you also have the reasons for refusal?
- Chapter 2.2: I would like to know how you processed the data. Did you collect the screening test results, for example, in an Excel sheet or some specific database?
- Chapter 2.2.2 lines 136-141: Here criteria for failing and passing the test are given. It seems that, irrespective of the result of the TEAOE test, the AABR test result was the one that decided whether the test was passed or failed. In the remaining manuscript, the TEOAE test is not further mentioned. I am wondering why TEOAE testing was performed if this is not important for the screening test outcome, and the difference in the performance of both tests was not compared in the manuscript.
- Chapter 2.2.3: Could you give more details on the sample size calculation? Was this done to reach a 95% confidence interval, which goes, for example, from 1% to 6% (your stated “5% precision”)? The obtained numbers seem pretty low for that. Did you calculate the sample size for the Clopper-Pearson confidence interval, and what software did you use? Did you also stop your study after the required sample size was reached, or did you calculate the sample size after the study?
- Chapter 2.3. Data analysis: In my opinion, the authors have not given enough information on how the statistical analysis was performed. In more detail, they state that a chi-square test was used to determine the association between the categorical variable. However, continuous variables were analyzed (age, gestational age,…). What tests were used for such variables, and did you assume normal distribution? It is great that the type of confidence interval is given for the proportion of neonates with “hearing loss” (which, in my opinion, is the screening failure rate).
Afterwards, univariate and multivariable logistic regression were used to investigate risk factors for hearing loss. What was the outcome here? For each participant, “yes/no” for hearing loss, and you chose the “yes” category for the modelling, and “no” was the reference category? Also, according to the STROBE checklist, all outcomes and predictors, and potential confounders should be clearly defined.
It seems that not all variables from Table 1 were included in the regression analyses. It should be stated which variables were excluded and why.

5) Results:
- Lines 161-162: The results are stated with two digits, which implies a higher precision than what is possible. With a total sample size of 60, I would suggest reducing the number of decimal places to 1. This applies to all results in the manuscript. For example, the gestational age of 36.37 weeks could be changed to 36.4 weeks (SD = 3.4).
- Line 164: A reference to Figure 1 is made, where the birth weight has been categorized, and also the gestational weeks. In the regression analysis, you work with the continuous birth weight and gestational age variables. I would suggest improving Figure 1 and removing the categorizations as stated below.
- Lines 170-171: Here, the hearing loss of over 28% is reported (as 43/60 by the way and not as 17/60). In my opinion, this is the test failure rate rather than the hearing loss rate.
- Lines 170-181: Please give a reference to your statements (probably Table 1).
- Lines 172-175: Please consider adding units to the results.
- Lines 182-187: The logistic regression results are presented for hearing loss as an outcome. I have great concerns about the results. For example, the odds ratio for gestational age is presented as 1.85. This would mean that for a week increase in gestational age, the odds for hearing loss increase by 1.85. However, in Table 1, it can be seen that the birth weight was statistically significantly higher in the group of neonates who passed the screening. This is contradictory. Another example: Premature babies have decreased odds for hearing loss (OR = 0.62) compared to babies who were not premature. In Table 1, it can be seen that the proportion of premature neonates in the “failure” group was much higher than in the group of neonates who passed the screening. I recommend revising the logistic regression analyses.
Also, the odds ratios for birth weight are weird. The authors might want to check the unit of the variable and consider transforming it (for example, per 10 gram/100 gram) to yield meaningful odds ratios. If there is only a little variation in the birth weight, it does not make sense to model a change of 1 kg in it.

6) Discussion
- The statements regarding the own study results should be rewritten after correcting the analyses.

7) Figures and Tables:
- Figure 1: Figure 1 is really hard to read. It seems to be made by ggplot in R. I would suggest showing boxplots with added individual data instead of bar plots showing relative frequencies. The disadvantage here is that you do not see the absolute numbers, and the continuous variables are split into categories (which are not used in that way in the analyses). With boxplots, one could see how the continuous variables are distributed without the need for categorization. Such plots can also easily be made in ggplot. You might also want to consider a “2 column 3 row” design instead of “3 column 2 row” to enhance readability. Suggestion: You could make birth weight vs. gender (A), gestational age vs. gender (B), birth weight vs. screening test result (C), gestational age vs. test result (D), birth weight vs. mode of delivery (E), and gestational age vs. mode of delivery (F).
- Table 1: Please consider revising the title because it says hearing loss. The row “hearing test” could be changed to failed hearing test or similar. Birth weight and mother’s age are shown without units. Please also consider adding explanations to the used abbreviations, such as CI and SD.
- Table 2: Please consider revising the title because it says hearing loss, adding units and explanations to the used abbreviations, and calculating the results again.
- Both tables: If you use tbl_summary of the gtsummary package in R, could you automatically highlight statistically significant p values using “bold_p”?

8) Supplementary materials:
- The STROBE checklist is lacking all relevant text from the manuscript, which should be included for each item. This would also help the authors to see if they have included everything according to the checklist. Additionally, some answers are missing (items 12c, 13 b, 13c, 14 b, 16 b, 16c).

Reviewer 2 ·

Basic reporting

Yikawe et al have submitted a very interesting paper on hearing screening from Nigeria. This paper describes failed newborn hearing screening in a nursery in Nigeria.

The study is well done, and the information is interesting and important.

Experimental design

Adequate. Informed consent is described, methods are well described, and conclusions are supported by the data.

Validity of the findings

There is no reason to question the validity of the findings. The findings are important, and clearly more follow-up is needed for these very vulnerable infants.

1. There were 60 neonates enrolled in the study. 57% were males and 43% were females, and their ages ranged from one to 30 days. A total of 44 (73%) neonates passed both hearing tests, while 16 (27%) failed. Factors such as low birth weight, prematurity, and birth asphyxia significantly correlated with the failure rates in these screenings (p<0.005).

Several questions arise:

i What were the percentages of “normal newborns” as opposed to “sick babies” (premature infants, LBW, birth asphyxia), and were the “referral rates” different in the two populations?

ii. Can the authors explain why boys were more likely than girls to get hearing screening?

iii. The “refer” rate of 27% seems high. How does this rate compare to similar populations of newborns? Do the authors have any explanation for this?

2. One other point should be clarified. It’s stated in the paper repeatedly that the study quantified hearing loss. But is this really true? In pediatric audiology, we draw distinctions between a “refer” on the newborn hearing screen and bona fide established hearing loss, which is established in follow-up studies.

Can the authors offer a reason why the paper should call this “hearing loss”, as opposed to “refer on the hearing screen”, which seems more accurate?

Additional comments

Some discussion on the most likely etiologies of hearing loss in Nigeria would add to the educational quality of the paper.

Reviewer 3 ·

Basic reporting

No comment

Experimental design

No comment

Validity of the findings

No comment

Additional comments

See the attached PDF

Annotated reviews are not available for download in order to protect the identity of reviewers who chose to remain anonymous.

---

## Round 0.2 · Major Revisions

Please revise following the reviewer's advice sincerely.

·

Basic reporting

no comment

Experimental design

no comment

Validity of the findings

All underlying data have been provided; they are robust, statistically sound, & controlled.
Conclusions still refer to hearing loss in one sentence, even if the study now assess risk factors for the hearing test referral rate. The rest of the conclusions seem to be fine.

However, the logistic regression analysis seems not be corrected and still shows unplausible results. Additionally, the odds ratios were not interpreted correctly. For this reason, the findings as stated should be checked.

Additional comments

Thank you for the second opportunity to review this interesting work entitled “Neonatal Hearing Screening Among High-Risk Newborns in Northwestern Nigeria”. The authors have greatly improved the manuscript based on the previous comments, which I really appreciate.
However, there are two main points left, which in my opinion still need clarification, and some minor points.

First major point: Logistic regression analyses
As I mentioned in my previous comments, there are problems regarding the regression analysis. However, I did not receive a response from the authors. I still believe that the logistic regression analyses were not performed correctly. Please allow me to insert my previous comments regarding that here:
“Lines 182-187: The logistic regression results are presented for hearing loss as outcome. I have great concerns about the results. For example, the odds ratio for gestational age is presented as 1.85. This would mean that for one week increase in gestational age, the odds for hearing loss increase by 1.85. However, in Table 1 it can be seen that the birth weight was statistically significantly higher in the group of neonates who passed the screening. This is contradictory. Another example: Premature babies have decreased odds for hearing loss (OR = 0.62) compared to babies who were not premature. In Table 1 is can be seen that the proportion of premature neonates in the “failure” group was much higher than in the group of neonates who passed the screening. I recommend to revise the logistic regression analyses. Also, the odds ratios for birth weight are weird. The authors might want to check the unit of the variable and consider to transform it (for example per 10 gram/100 gram) to yield meaningful odds ratios. If there is only a little variation in the birth weight it does not make sense to model a change of 1 kg in it.”
Using the included raw data, I have tried to reproduce the results of your logistic regression analyses. I could easily reproduce the results in Table 1. However, the regression analyses yielded different results. I would strongly recommend that you check which outcome you modeled. Please pay attention to the reference category in your outcome, which should be the babies who passed the test. Could it be that you modeled the outcome “passed the test” instead of “failed the test”? That is why I asked about it in detail. Checking the R code would help here.
I appreciate your interpretation of the odds ratios on lines 232-241. However, these interpretations do not seem to be correct. For example the OR of 1.85 (CI: 1.10-4.71) for gestational age (in weeks, modeled as a continuous variable), does not translate to (see lines 232-234 ): “Neonates born prematurely (i.e., with lower gestational age) were nearly twice as likely to fail the hearing screening compared to those born at term (OR = 1.85; 95% CI: 1.10-4.71).” The correct interpretation would be: Per one week increase in gestational age the risk of failing the hearing screening increases by a factor of 1.85 (95% CI: 1.10-4.71). This association is in the wrong direction. Hence I assume that you might not have modeled the risk of “failing the hearing screening” but the “risk of passing the hearing screening”. You can also see this if you use your variable “premature”, where the interpretation would be: Neonates born prematurely had reduced odds for failing the hearing screening compared to those born at term (OR = 0.62; 95% CI: 0.02-24.0). This association is also not in the correct direction.
Using the provided data I obtained an univariable OR of < 1 for gestational age in weeks and univariable OR >>1 for premature babies vs. babies born at term.
In your new analysis you should be able to find more statistically significant associations between failing the hearing screening and your study populations’ characteristics. You can basically orient yourself on the results in Table 1 (what is significant there, should also be significant or close to it in the regression, too, but in the “right” direction).
Another issue in my opinion is the multivariable analysis. I get the idea that you wanted to adjust for potential confounders and included all variables from the univariable analysis in the multivariable model. However, this comes with problems: You cannot include highly correlated variables in the same model. The best example is the gestational age in weeks and the birth weight of the baby. The variables are strongly correlated (Pearson correlation coefficient of approximately 0.7). It is a standard practice in such cases that you include only one of the both variables. If you still include both, the results will be biased (and in your case according to the raw data provided with the manuscript, in the multivariable model, increasing birth weight becomes a risk factor for hearing screening failure instead of being a protective factor like in the univariable model (based on my calculations)).
The same applies to the variable “premature”, which you created from the gestational age variable. Both variables should not be included in the same multivariable model.
I would recommend to carefully think what kind of model you want your adjusted model to be, I mean what factors you want to adjust for. Would adjusting for age and sex only be sufficient? In any case, please avoid including variables that are strongly correlated or created from each other.

Second major point: The sample size calculation
Thank you very much for providing further insights into the sample size calculation. However, I still believe that achieving a “5% precision” for a confidence interval of an assumed proportion of 2% (0.02) or 4% (0.04) with a sample size of 31 or 60 participants is difficult. I would really like to see the actual R command you used to calculate the sample size (together with the name of the R package), since the result sounds unusual. I would like to understand the sample size calculation more deeply.
Also, why did you base your sample size calculations on an outcome for hearing loss when your objective was to assess risk factors for failed screening tests?
In the revised manuscript, you mention that the sample size calculation aligns with that of similar single-center studies, such as in Oyinwola (2023). However, in that study, the authors included a total of 160 participants and did not provide much details on how the sample size was arrived at.



The following minor points are structured along the manuscript from the abstract to conclusion.

Abstract:
1. The odds ratios seem to be not correct, see also my first major comment. I recommend running the logistic regression analyses again and reporting the new odds ratios in the abstract.

Introduction:
2. Line 61: The abbreviations NICU and LMIC are introduced here but not explained (LMIC also not later in the manuscript). You might want to introduce the abbreviations here.
3. Line 77: if you abbreviated NICU on line 61, you can remove the definition from here.
4. Line 83: SNHL was already defined on lines 63-64 and can be removed from here.

Materials and methods:
5. Line 119: NICU has been already previously defined.
6. Line 123-124: In your previous comments you stated that 60 out of 150 neonates were recruited in the study, which would be a participation rate of 40%. I am wondering if this rate might be good to include in the manuscript? You reflect on this on the lines 282-284 and could maybe support this by the actual number of the participation rate, if you wish. Future studies could then directly cite that number, which might be a pretty good participation rate in the NICU setting in Nigeria (?).
7. Line 133: Which R Version was used? Here you state R Version 4.3.3. and in the abstract R Version 4.0.3. Or was is both?

Results
8. Please consider revising the regression analyses and reporting the new results and interpreting them.

Conclusion
9. Lines 295-297: Here you refer to factors of hearing loss, but in the new version the topic is about hearing screening failure. I recommend rephrasing this sentence.

Figures and Table:

10. Table 1: Please consider revising the title because it says hearing loss. Birth weight and mother’s age are shown without unit. You could also add the explanation to the abbreviation “SD”.
11. Table 2: Please consider revising the title because it says hearing loss, adding units and explanations to the used abbreviations, and calculating the results again.

---

## Round 0.3 · accepted · Accept

Congratulations, we are glad to accept your paper.

·

Basic reporting

no comment

Experimental design

no comment

Validity of the findings

no comment

Additional comments

Thank you very much for addressing all my comments and revising the regression analyses.